# Phafins Are More Than Phosphoinositide-Binding Proteins

**DOI:** 10.3390/ijms24098096

**Published:** 2023-04-30

**Authors:** Tuoxian Tang, Mahmudul Hasan, Daniel G. S. Capelluto

**Affiliations:** 1Department of Biology, University of Pennsylvania, Philadelphia, PA 19104, USA; 2Protein Signaling Domains Laboratory, Department of Biological Sciences, Fralin Life Sciences Institute and Center for Soft Matter and Biological Physics, Virginia Tech, Blacksburg, VA 24061, USA

**Keywords:** Phafin, PH domain, FYVE domain, PtdIns(3)P, PtdIns(4)P, autoinhibition, membrane remodeling

## Abstract

Phafins are PH (Pleckstrin Homology) and FYVE (Fab1, YOTB, Vac1, and EEA1) domain-containing proteins. The Phafin protein family is classified into two groups based on their sequence homology and functional similarity: Phafin1 and Phafin2. This protein family is unique because both the PH and FYVE domains bind to phosphatidylinositol 3-phosphate [PtdIns(3)P], a phosphoinositide primarily found in endosomal and lysosomal membranes. Phafin proteins act as PtdIns(3)P effectors in apoptosis, endocytic cargo trafficking, and autophagy. Additionally, Phafin2 is recruited to macropinocytic compartments through coincidence detection of PtdIns(3)P and PtdIns(4)P. Membrane-associated Phafins serve as adaptor proteins that recruit other binding partners. In addition to the phosphoinositide-binding domains, Phafin proteins present a poly aspartic acid motif that regulates membrane binding specificity. In this review, we summarize the involvement of Phafins in several cellular pathways and their potential physiological functions while highlighting the similarities and differences between Phafin1 and Phafin2. Besides, we discuss research perspectives for Phafins.

## 1. Introduction

Living cells utilize endocytosis to take up extracellular nutrients and membrane-bound macromolecules from the plasma membrane and transport them to the cytoplasm in endocytic vesicles, where the cargo is internalized and invaginated into vesicular and tubular carriers [1,2,3]. When cells are in a nutrient-depleted state, they exploit an evolutionarily conserved process, known as autophagy (“self-eating”), to survive [4,5,6]. The plasma membrane and membrane-enclosed organelles, such as endosomes, the endoplasmic reticulum (ER), lysosomes, and the Golgi apparatus, undergo fusion, fission, and maturation processes to coordinate cargo transportation. These dynamic cellular membrane rearrangement processes are collectively referred to as “membrane dynamics” [7,8,9].

Membrane dynamics depends upon the presence of rare phospholipids known as phosphoinositides (PIPs) that reversibly recruit cytosolic proteins via their conserved PIP-binding domains to specific cellular membranes in a spatiotemporally controlled fashion. PtdIns(3)P, PtdIns(4)P, PtdIns(5)P, PtdIns(3,4)P_2_, PtdIns(3,5)P_2_, PtdIns(4,5)P_2_, and PtdIns(3,4,5)P_3_ are the seven PIPs found in animal cells. PIPs serve as identity markers of membranous compartments. For example, the plasma membrane is highly enriched in PtdIns(4,5)P_2_, whereas PtdIns(4)P and PtdIns(3)P predominate in the Golgi apparatus and the early endosomes, respectively [10,11]. Multiple PIP kinases and phosphatases are responsible for the fast turnover of PIP and the versatile and dynamic conversion of PIPs makes them optimal regulators of signal transduction and membrane remodeling [12,13].

The Phafin protein family is categorized into two groups: Phafin1 and Phafin2. These proteins are composed of two PIP-binding domains: the PH and FYVE domains. Phafin1 and Phafin2 preferentially bind to PtdIns(3)P and weakly bind to PtdIns(4)P and PtdIns(5)P [14,15,16]. Structural analysis shows that Phafin2 is an α/β protein with ~40% random coil content [17]. An autoinhibition mechanism for PtdIns(3)P binding has been reported in Phafin2, by which its conserved C-terminal aspartic acid-rich (polyD) motif intramolecularly inhibits the PH domain binding to PtdIns(3)P [16]. The C-terminal tail of Phafin1 provides a lysosomal targeting signature and autophagy induction signals [15]. A recent bioinformatics study showed that the PH domain and the C-terminal polyD motif of Phafin2 exhibit a unique concurrence in animals [18]. All these pieces of evidence suggest that the polyD motif may be required as a molecular mechanism to control the membrane targeting of Phafin proteins.

## 2. The Phafin Protein Family

### 2.1. Overview of Phafin Protein Family

Members of the Phafin protein family are Phafin1 and Phafin2. Phafin1, also known as LAPF (a lysosome-associated apoptosis-inducing protein containing the PH and FYVE domains) and PLEKHF1 (pleckstrin homology and FYVE domain containing 1), is a 279 amino acid protein. Genomic data shows that the human *Phafin1* gene is located on chromosome 19q12 (https://www.ncbi.nlm.nih.gov/gene/79156 (accessed on 9 February 2021)). RNA-seq experiments performed on tissue samples, representing 27 different tissues from 95 human individuals, revealed that the *Phafin1* gene is ubiquitously expressed in fat, spleen, and, to a lesser extent, other tissues [19].

Phafin2, also known as EAPF (an endoplasmic reticulum-associated apoptosis-involved protein containing PH and FYVE domains) and PLEKHF2 (pleckstrin homology and FYVE domain containing 2), is a 249 amino acid protein [17,20]. The human *Phafin2* gene is located on chromosome 8q22 (https://www.ncbi.nlm.nih.gov/gene/79666 (accessed on 9 February 2021)). Quantitative transcriptomics analysis showed that the human *Phafin2* gene is broadly expressed in bone marrow, lymph nodes, and other tissues [19]. The overall expression level of human Phafin2 is higher than Phafin1 in different tissues.

Structurally, Phafin1 and Phafin2 share a similar modular organization, containing an N-terminal PH domain, a central FYVE domain, and a polyD motif (Figure 1). A major difference between Phafin1 and Phafin2 is that the Phafin1 polyD motif is shorter and is followed by a C-terminal tail. Functionally, both Phafin1 and Phafin2 proteins have been reported to play significant roles in apoptosis and autophagy. Phafin2 promotes macropinocytosis through the rearrangement of the subcortical actin cytoskeleton and recruits JNK (c-Jun NH2-terminal kinase)-interacting protein 4 (JIP4) to tubulating macropinosomes [21,22]; whereas the role of Phafin1 in macropinocytosis remains to be determined.

To carry out their cellular functions as adaptor proteins, cytosolic Phafins are associated with intracellular membranes and recruit their binding partners. Membrane association is mediated through direct binding to PIPs as Phafin proteins are equipped with two PIP-binding domains: the PH and FYVE domains.

### 2.2. The Availability of Phafins among Animal Species

The presence of Phafins is ubiquitous across the animal kingdom. To explore orthologues, paralogues, or transcript variants of human genes in different animal species sets, ‘Ensembl Human Genome Browser 109’ serves as a useful database [23]. The database has enlisted the presence of homologous orthologues of human Phafin1 in 132 species among 199 of total species sets. The reported orthologues include 19 primate species; 2 rodent species; 30 species of carnivores, ungulates, and insectivores; 72 mammalian species; 26 bird and reptile species; and 25 fish species. On the other hand, the ‘Ensembl Human Genome Browser’ has reported approximately 202 human Phafin2 homologous orthologues. These orthologues have been found in 21 primate species; 19 rodent species; 35 species of carnivores, ungulates, and insectivores; 79 mammalian species; 25 bird and reptile species; and 56 fish species. The abundance of Phafin2 proteins in mammalian species, as evidenced by the highest number of Phafin2 orthologues in the database, suggests that this protein is more prevalent in mammals compared to other species sets. We previously observed that all mammalian homologues of Phafin2 contain the C-terminal acidic polyD region in addition to the PH and FYVE domains [18]. Additionally, the study incorporated various nonmammalian model organisms, including Arabidopsis thaliana, Saccharomyces cerevisiae, Danio rerio, Drosophila melanogaster, Xenopus tropicalis, and Caenorhabditis elegans. Interestingly, the domain-based search found that the protein encoding a polyD motif along with the PH-FYVE domains was solely detected in zebrafishes. In addition, despite both PH and FYVE domains are found individually in Arabidopsis proteins, the polyD motif is missing not only in this organism but also in plants in general [18].

## 3. PIPs and PIP-Binding Domains

PIPs are phosphorylated derivatives of phosphatidylinositol (PtdIns). The *myo*-inositol headgroup of phosphatidylinositol can be phosphorylated at a single position or a combination of positions (3’, 4’, or 5’), producing seven different PIPs: PtdIns(3)P; PtdIns(4)P; PtdIns(5)P; PtdIns(3,4)P_2_; PtdIns(3,5)P_2_; PtdIns(4,5)P_2_; and PtdIns(3,4,5)P_3_ (Figure 2A) [24,25,26,27]. Although PIPs represent a minor group of phospholipids (<10% of total phospholipids) and comprise approximately 1% of cellular lipids, they are key signaling molecules. PIPs regulate multiple cellular functions, including membrane trafficking, signaling transduction, cell survival, and cytoskeletal dynamics [28,29,30,31].

The amount and cellular distribution of each PIP are spatially and temporally controlled by numerous kinases and phosphatases, which constitute a complicated regulatory network. PtdIns(4)P and PtdIns(4,5)P_2_ are the largest pools of PIPs. They are constitutively present in cellular membranes, whereas PtdIns(3,4,5)P_3_ is barely detectable in unstimulated cells. Recent estimates have shown that the amount of PtdIns(3,4,5)P_3_ is 2–5% of PtdIns(4,5)P_2_ and PtdIns(3)P is about 20–30% of PtdIns(4)P [27]. Different PIP species show distinct cellular distributions, and some are viewed as hallmarks of various membrane compartments. PtdIns(4)P is abundant at Golgi apparatus membranes, whereas PtdIns(3)P predominates in early and late endosomes and lysosomes [32,33,34,35].

PIPs serve as site-specific signals and docking sites on cellular membranes. Some cytosolic proteins are transiently recruited to a specific membrane by these phospholipids in a reversible and regulated manner [36,37,38]. Membrane recognition is mediated by PIP-binding domains, including ANTH (AP180 N-terminal homology) [39], BAR (Bin-Amphiphysin-Rvs) [40], C2 (conserved region-2 of protein kinase C) [41], ENTH (epsin N-terminal homology) [42], FYVE (Fab1, YOTB, Vac1, and EEA1) [43], PH (pleckstrin homology) [44], PROPPINs (β-propellers that bind to PIs) [45], PTB (phosphotyrosine-binding) [46], and PX (phox homology) domains [47]. PIP-binding domains are diverse in their specificity. Some specifically bind to one type of PIP, while others display broader specificity. For example, FYVE domains are specific for PtdIns(3)P, whereas PH domains bind to nearly all phosphoinositides [48,49].

Although PIP-binding domains display distinct structures and affinities, their lipid-binding mechanisms share many similarities. In most cases, binding is achieved by nonspecific electrostatic interactions involving negatively charged PIP headgroups and patches of positive charges on the protein surface [50,51]. Nearly all PIP-binding domains have highly basic binding sites. Furthermore, the association of PIP-binding proteins with PIP-containing membranes may involve multiple interactions. In addition to electrostatic interactions, hydrophobic amino acid residues located near the basic patch contribute to lipid ligation. Membrane anchoring is also augmented by additional mechanisms including hydrophobic insertion, protonation of a histidine switch, and coincidence detection [52,53,54,55].

## 4. The PIP-Binding Domains of Phafin Proteins

### 4.1. The PH Domain

The PH domain was initially identified as a region of sequence homology duplicated in Pleckstrin [56]. PH domains are small protein modules of 100–120 amino acids found in a wide range of proteins [57]. The amino acid sequence homology among different PH domains is only 7–30%, but they show similar three-dimensional structures. PH domains are characterized by the presence of seven β-strands, forming two nearly orthogonal antiparallel β-sheets that are capped at one end by a C-terminal α-helix, as is the case for the structural model of the Phafin2 PH domain (Figure 2B) [58,59,60,61].

Several PH domains bind to PIPs, facilitating the recruitment of PH domain-containing proteins to specific cellular membranes. A positive binding pocket in PH domains interacts with the negative head group of the phospholipids and nonspecific electrostatic interactions are the major driving force for binding. It is estimated that about 15% of PH domains bind to PIPs with high affinity, but dissociation constants range between 30 nM to 30 μM [57]. The specificity of the PH domain varies widely. For example, the PH domains of Pleckstrin [62] and PLCδ1 [63] bind to PtdIns(4,5)P2; the Btk [64] and Grp1 [65] PH domains have specificity for PtdIns(3,4,5)P3; and the PH domain of Akt/PKB binds to both PtdIns(3,4)P2 and PtdIns(3,4,5)P3 [66,67]. Moreover, PH domains also bind to PtdIns(3)P [68]; PtdIns(4)P [69]; PtdIns(5)P [70]; and PtdIns(3,5)P2 [71]. The Phafin2 PH domain preferentially binds to PtdIns(3)P, but it also weakly binds to PtdIns(4)P and PtdIns(5)P [21,72].

A highly basic sequence motif (KXn(K/R)XR) located in the β1/β2 loop can be found in many PH domains and is proposed to be a contact site between the PH domain and membrane PIP [55,73]. This putative PIP-binding motif is represented by the sequence 49-KAKPR-53 in human Phafin1 or the sequence 49-KPKAR-53 in human Phafin2 (Figure 1 and Figure 2B). Indeed, the R53C mutation in the Phafin2 PH domain significantly decreased PtdIns(3)P-binding [16]. Recent studies have reported that multiple PIP-binding sites are present in PH domains and that PIP-binding occurs in regions of PH domains other than the classic binding motif [55,73]. It is unclear whether Phafin PH domains bear noncanonical PIP-binding sites. Since Phafin2 is a coincidence detector of PtdIns(3)P and PtdIns(4)P in the macropinocytic pathway [21], it remains an open question as to whether PtdIns(3)P and PtdIns(4)P overlap their binding sites in the PH domain.

The high affinity and specificity of some PH domains make them suitable probes for detecting a specific PIP in vivo. Fusion proteins comprising green fluorescent proteins (GFPs) and PH domain have become useful and efficient tools to study the distribution of PIPs in living cells. For example, the PH domain of Akt/PKB is a sensitive and selective biosensor for detecting cellular PtdIns(3,4)P2 and PtdIns(3,4,5)P3 [74], whereas cellular PtdIns(4,5)P2 can be detected by the use of the PH domain of PLC-δ1 [74,75,76].

In addition to lipid binding, PH domains also mediate protein–protein interactions [77]. Previous studies showed that the Btk PH domain interacts with protein kinase C (PKC) and the Akt PH domain forms a tight complex with calmodulin [78,79]. The Phafin2 PH domain directly binds F-actin and JIP4 (Table 1) [21,22]. The presence of both the PH and FYVE domains in Phafin proteins enables their interactions with membrane-embedded lipids and other peripheral proteins, which characterize them as adaptor proteins; however, it remains unknown whether any of these associations and PIP-binding are mutually exclusive.

### 4.2. The FYVE Domain

The FYVE domain is named after the first letter of the four proteins, Fab1, YOTB, Vac1, and EEA1, in which it was first identified. FYVE domains are small (70–80 amino acids), cysteine-rich protein modules that are highly conserved, from yeast to humans. FYVE domain-containing proteins are recruited to PtdIns(3)P-enriched endocytic membranes through specific interactions of their FYVE domains with PtdIns(3)P [82,83,84]. The high specificity and affinity of FYVE domains for PtdIns(3)P-binding make them ideal PtdIns(3)P detectors. For example, the double FYVE finger (2×FYVE) is widely used to study the distribution and level of PtdIns(3)P in eukaryotic cells [85,86,87].

Three-dimensional structures of FYVE domains reveal a common protein fold. They display two double-stranded β-sheets, with each β-sheet formed by two antiparallel β-strands as predicted for the Phafin2 FYVE domain (Figure 2C). The structure contains a small C-terminal α-helix that carries one of the zinc-coordinating cysteines. There are three conserved PtdIns(3)P-binding motifs in FYVE domains: an N-terminal WxxD (in single-letter amino acid code; x, any amino acid) motif, a central (R/K)(R/K)HHCR motif, and a C-terminal RVC motif (Figure 1 and Figure 2C) [88,89,90,91]. Certain FYVE domains bind PtdIns(3)P in a pH-dependent manner. An acidic pH (~6.0) favors PtdIns(3)P-binding, driven by two adjacent histidine residues in the (R/K)(R/K)HHCR motif [92,93]. In contrast, Phafin2’s PtdIns(3)P-binding is pH-independent [94].

The FYVE domain is a zinc-finger module. It contains eight cysteine residues (four CxxC motifs) that coordinate two Zn^2+^ ions. The tetrahedral coordination of two Zn^2+^ ions by four CxxC motifs is critical to the structural stability and biological activity of the FYVE domains. Furthermore, some FYVE domains promote protein dimerization [83]. Structures of the *Drosophila* Hrs FYVE and the *Saccharomyces cerevisiae* Vps 27p FYVE domains were determined by X-ray crystallography. The Hrs FYVE domain was crystallized in the presence of citrate (a substrate substitute), whereas the structure of the Vps 27p FYVE domain was solved in a ligand-free state [95]. These two FYVE domains had similar structures comprised of two double-stranded β-sheets and a C-terminal α-helix. According to the proposed model, two Hrs FYVE domains form a homodimer, but the Vps 27p FYVE domain forms a monomeric structure. The crystal structure of the human EEA1 FYVE domain bound to inositol 1,3-bisphosphate also revealed that EEA1 FYVE domains form homodimers [88,89,96]. Phafin2 FYVE is likely to be monomeric as Phafin2 is a moderately elongated monomer in a free state [17].

## 5. The Functions of Phafin Proteins in the Apoptotic, Autophagic, and Endocytic Pathways

### 5.1. Apoptosis

Apoptosis is a type of programmed cell death that is characterized by various morphological changes, including cell shrinkage, chromatin condensation, plasma membrane blebbing, and apoptotic bodies. The prime physiological role of apoptosis is to maintain cellular homeostasis by eliminating dying cells [97]. Additionally, the neurological and immune systems develop through excessive production of cells, and redundant cells are cleared away by apoptosis as they fail to present productive antigen specificities or synaptic connections [98]. Alteration of this balanced homeostasis induces abnormal cellular growth leading to deadly diseases including cancer development.

There are two main apoptotic pathways: the mitochondrial (intrinsic) and the death receptor (extrinsic) pathways [99,100]. Mitochondria-initiated events happen in the intrinsic apoptotic signaling pathway where non-cell death receptor-mediated stimuli can initiate intracellular signals to specific cell targets for cell-required consequences [101]. In contrast, cell-surface death receptors initiate the extrinsic apoptotic pathways [102,103]. Tumor necrosis factor-α (TNF-α) can initiate the extrinsic death receptor pathway directly and the mitochondrial pathway indirectly.

Phafin2 promotes TNF-α-induced cellular apoptosis through an endoplasmic reticulum (ER)–mitochondrial apoptotic pathway [20]. Overexpression of Phafin2 enhances cellular sensitivity to apoptosis induction and TNF-α-induced activity of caspase 3. Both the Phafin2 PH and FYVE domains contribute to the translocation of Phafin2 to the ER leading to an increase in cytosolic Ca^2+^ levels and, simultaneously, causes a reduction in protein unfolding responses in this compartment [20,104]. Deletion of either the PH or FYVE domain impairs Phafin2 and causes it to colocalize with the ER after TNF-α treatment, suggesting that both domains are required for ER localization. Moreover, these two Phafin2-deletion mutants reduce the sensitivity of L929 cells to TNF-α-mediated apoptosis [20].

### 5.2. Endosomal Cargo Transport

The Phafin PH and FYVE domains may play a distinct role in endosomal membrane association despite both binding to PtdIns(3)P [16,72]. Phafin2 that lacks its PH domain still binds to endosomes, whereas removal of its FYVE domain blocks endosome-binding, indicating that the FYVE domain is indispensable for Phafin2 endosomal targeting. It has been reported that Phafin2 modulates the structure and function of endosomes [14]. High levels of Phafin2 in both HeLa and HEK293T cells lead to enlarged endosomes [14].

The Rush hour protein, a *Drosophila* homolog of human Phafin1 and Phafin2 (Figure 1), regulates endosomal trafficking from early endosomes to late endosomes and from late endosomes to lysosomes by modulating the activity of Rab proteins. The Rush hour protein localizes to endosomes through interactions between the Rush FYVE domain and endosomal PtdIns(3)P. Similar to that described in mammalian cells, overexpression of Rush results in enlarged endosomes and disrupts endocytic cargo progression. Rush directly binds to Rab GDP dissociation inhibitor (GDI), which is recruited to endosomal membranes (Table 1). The increased level of Rush may cause entrapment of GDI at endosomal membranes, giving rise to over-activation of Rab proteins [80].

Phafin2 modulates the presence of several receptors on the cell surface by regulating receptor internalization/recycling [105]. Overexpression of Phafin2 increases the density of insulin receptor (InsR), interferon receptor, interleukin-6 receptor, purinergic receptor P2Y5, and transforming growth factor-beta receptor, but does not change the levels of either the hepatocyte growth factor receptor or the epidermal growth factor receptor (EGFR) on the cell’s surface [14]. Downregulation of Phafin2 expression decreases the level of InsRs on the cellular membrane. In contrast to control or Phafin2-depleted cells, InsR immunocomplexes are retained at the plasma membrane in Phafin2-transfected cells, indicating that a high level of Phafin2 disrupts the internalization of InsRs, promoting accumulation of InsRs on the cell surface. An increased density of InsRs amplifies insulin stimulation of cells and upregulates the activity of the activator protein 1, which is a downstream mediator of the insulin pathway [14].

In contrast to that which is observed in [14], Phafin2 was reported to facilitate degradation of epidermal growth factor (EGF) and EGFR. Knocking down Phafin2 precludes the transport of the EGF–EGFR complex through early endosomes. In Phafin2-depleted cells, the average size of early endosomes is significantly decreased, which promotes a delay of EGFR transport in these compartments [81]. Thus, Phafin2 may regulate EGFR degradation by mediating endosome fusion. Through the yeast two-hybrid assay, it was shown that Phafin2 colocalizes with and binds to the early endosomal antigen 1 (EEA1) (Table 1), an endosomal-tethering protein [81]. EEA1 colocalizes and physically interacts with Rab5 on early endosomes, regulating early endosome fusion [106]. This suggests that Phafin2 regulates endosome fusion via association with EEA1. Targeted siRNA screening for PtdIns(3)P-binding proteins involved in EGFR degradation found that Phafin1 also plays a role in the degradation of EGFR [81], but the underlying mechanism remains unexplored.

Overexpression of GFP-tagged Phafin1 in HEK 293T cells causes the formation of enlarged endosomes, suggesting that Phafin1 may be involved in the modulation of endocytosis. Additionally, Phafin1 is recruited to lysosomes in a Rab7-dependent manner, inducing autophagy [15]. Rab7 is a RAS-related GTP-binding protein localized to late endosomes, while Rab5 is primarily localized to early endosomes [107]. Therefore, Phafin1 and Phafin2 may carry out their functions in distinct phases of endosomal/lysosomal pathways with Phafin2 having a role in early endosomal fusion and Phafin1 contributing to late endosomal fusion [81].

### 5.3. Autophagy

Autophagy is a highly conserved process occurring in eukaryotic cells, by which damaged cytoplasmic components, including misfolded proteins and defective organelles, are sequestered in double-membrane vesicles, known as autophagosomes, and delivered to lysosomes for degradation [108,109,110,111]. Metabolites produced by this degradation process can be recycled to build new cellular macromolecules and functional organelles. Under normal growth conditions, autophagy is active at basal levels, maintaining cellular homeostasis; however, cellular stress, such as nutrient starvation, growth factor deprivation, infection, and accumulation of misfolded proteins, may induce autophagy, which can be selective (e.g., protein aggregates) or nonselective (sequestration of bulk cytoplasm), depending on the stimulating factor [112,113].

Based on their different mechanisms and functions, there are three primary types of autophagy: microautophagy, macroautophagy, and chaperone-mediated autophagy [114,115]. In microautophagy, cytoplasmic components are directly engulfed by lysosomes through the invagination of the lysosomal membrane. In the case of macroautophagy (often referred to as autophagy), a unique membrane structure, called the isolation membrane or phagophore, emerges after autophagy induction. The isolation membrane expands and elongates to form autophagosomes, which are double-membrane vesicles that engulf cytosolic components. Autophagosomes fuse with lysosomes resulting in the formation of autolysosomes. Lysosomal hydrolytic enzymes digest the autolysosomal contents, and the resulting degradation products are transported back to the cytosol through membrane permeases (Figure 3A) [108]. Chaperone-mediated autophagy relies on the action of cytosolic and lysosomal Hsc70 chaperone, which recognizes the soluble cytosolic proteins carrying the pentapeptide KFERQ-like sequence [116]. The Hsc70 chaperone associates with the integral membrane receptor LAMP-2A (lysosome-associated membrane protein type 2A), which translocates substrate proteins across the lysosomal membrane [117]. Autophagy plays a key role in cellular adaptation to changing environmental conditions, cellular remodeling during development and differentiation, and lifespan determination [118]. Autophagic dysfunction is associated with a plethora of human diseases, such as cancer [119], neurodegeneration [120], microbial infection [121], myopathies [122], and aging [123].

Knowledge of autophagy has been markedly expanded after identifying more than 20 different *ATG* (autophagy-related) genes from yeast genetic studies. *ATG* genes encode evolutionarily conserved Atg proteins, which are essential to the autophagic machinery. For example, autophagosomal membrane expansion is mediated by two ubiquitin-like conjugation systems (Atg12 and Atg8). The ULK (Unc-51-like kinase) complex (Atg 1 (ULK1/2), Atg 11, Atg 13, Atg 17, Atg 29, and Atg 31) is indispensable to the induction of autophagy [124]. After autophagy induction, this complex translocates to the early autophagic machinery.

PtdIns(3)P is indispensable to autophagy induction, a process that depends on PI3K (phosphoinositide 3-kinase) activity [125,126]. PI3K phosphorylates the 3’-OH group of the inositol ring of phosphoinositides. Many PtdIns(3)P-binding proteins are involved in the autophagic process, such as DFCP1 (double FYVE-containing protein 1) and WIPI (WD repeat domain phosphoinositide-interacting) families of proteins, which are critical to autophagosome formation [109,110,111,112,113]. Phafin2 plays a role in the initiation of autophagy (Figure 3B) [72,127,128]. Lysosomal accumulation of Phafin2, in complex with the serine/threonine kinase Akt (also known as protein kinase B, PKB), constitutes a critical step in the induction of autophagy (Table 1). There are three highly conserved Akt isoforms in mammals: Akt1 (PKBα), Akt2 (PKBβ), and Akt3 (PKBγ). Akt proteins are involved in many cellular functions, including cell growth, cell survival, vesicular trafficking, transcriptional regulation, and cytoskeletal organization [129,130]. Akt1 and Akt2, but not Akt3, interact with Phafin2 in HEK293T cells. The PI3K-Akt-mTOR (mechanistic target of rapamycin) pathway plays an essential role in the regulation of autophagy. After induction of autophagy using rapamycin, Phafin2 is localized to lysosomes via interactions with PtdIns(3)P, thus, recruiting Akt to these compartments [72]. Both the Phafin2 PH and FYVE domains are required for the association of Phafin2 to Akt. Moreover, Phafin2 binds to Akt in a non-phosphorylation-dependent manner; both the phosphorylated and non-phosphorylated Akt bind to Phafin2 [72].

It is important to point out that Akt plays a dual role in the regulation of autophagy (Figure 3B) [128]. Akt has an N-terminal PH domain, a kinase domain in the middle, and a C-terminal regulatory domain. The Akt PH domain has an inhibitory effect on the Akt-kinase domain, and Akt is in the inactive “PH-in” conformation. The “PH-in” conformation is relieved after the Akt PH domain associates with PtdIns(3,4,5)P_3_, resulting in a “PH-out” conformation. The Akt-kinase domain is available for phosphorylation by PDK1 (phosphoinositide-dependent protein kinase 1). PDK1 phosphorylates Akt at T308, leading to a partially activated Akt. To be fully functional, Akt also requires S473 phosphorylation by mTORC2 (mechanistic target of rapamycin complex 2) [129]. In response to growth factor stimulation, class I PI3K is activated and increases the PtdIns(3,4,5)P_3_ levels at the plasma membrane. The Akt PH domain associates with PtdIns(3,4,5)P_3_, translocating Akt to the plasma membrane. The binding of Akt to PtdIns(3,4,5)P_3_ causes conformational changes in Akt, promoting its kinase activation [131]. Activated Akt can phosphorylate downstream substrates, leading to the inhibition of autophagy [129].

Phafin1 is also involved in autophagy [15]. Co-overexpression of Phafin1 and hLC3A is sufficient to induce autophagy. The Phafin1 C-terminal tail, which is absent in Phafin2, is indispensable for lysosomal targeting and autophagy activation functions [15]. These observations indicate that Phafin1 and Phafin2 do not have redundant cellular functions.

### 5.4. Macropinocytosis

Macropinocytosis is an actin-dependent endocytosis mechanism that cells exploit to ingest extracellular fluids and soluble macromolecules. Cargo uptake can be divided into two morphologically different steps: membrane ruffling and closure of membrane ruffles. Membrane ruffling requires rigorous remodeling of the plasma membrane followed by the internalization of extracellular fluids and solutes into cup-shaped vacuoles, which then mature into large vesicles, called macropinosomes [132,133]. Macropinocytosis is a transient cellular process in most cell types while some stimulating factors, such as EGFs and phosphatidylserine, promote this cellular activity [134]. Macropinocytosis is important in immune response because macrophages and dendritic cells utilize it to capture and sample antigens, causing their presentation to the major histocompatibility complex. Moreover, a variety of pathogens, such as bacteria and viruses, opportunistically take advantage of macropinocytosis to invade host cells [135,136].

A recent study demonstrated that Phafin2 promotes macropinocytosis by coordinating actin organization at macropinosomes that are forming [21]. Phafin2 is transiently recruited to the newly formed macropinosomes through coincidence detection of PtdIns(3)P and PtdIns(4)P (Figure 4A). Phafin2 shows two distinct phases of localization to the same macropinosome. The first phase of localization is a short-lived (~40 s), transient localization to the structures near the plasma membrane; the second phase of localization is a long-lasting (a few minutes) localization to large vesicles [21]. These two phases of localization represent two different steps in macropinosome maturation, namely nascent and early macropinosomes. The FYVE domain is required for the localization of Phafin2 to both nascent and early macropinosomes. On the other hand, the PH domain is only required for the localization of Phafin2 to nascent macropinosomes [21].

PtdIns(3)P binding is critical to the recruitment of Phafin2 to macropinosomes. The PtdIns(3)P-binding defective mutant Phafin2 R176A (within the FYVE domain) completely abolishes membrane association, highlighting the critical role of the FYVE domain in this process. Phafin2 relies on coincidence sensing of PtdIns(3)P by the FYVE domain and PtdIns(4)P by the PH domain during the first phase of localization to nascent macropinosomes [21]. The polyD motif, known to inhibit PH domain binding to PtdIns(3)P [16,137], prevents the nonspecific membrane binding of Phafin2. Newly formed macropinosomes are caught near the plasma membrane and are entangled in an actin matrix. These nascent macropinosomes are required to shed the dense actin coat on their limiting membrane to mature into early macropinosomes. Phafin2 modulates actin dynamics by binding to actin through its PH domain (Table 1), facilitating the entry of nascent macropinosomes through the subcortical actin matrix.

The motor-binding protein JIP4 is recruited to the tubulating macropinosomes by Phafin2 in a PtdIns(3)P-dependent manner (Figure 4B). JIP4, a coiled-coil protein that binds to dynein and kinesin motor protein complexes [22,138], directly interacts with the PH domain of Phafin2 (Table 1). Phafin2 dynamically colocalizes with JIP4 on early macropinosomes. Depletion of Phafin2 or JIP4 and disruption of PtdIns(3)P binding have a suppressive effect on membrane tubulation [22]. Interestingly, Phafin1 does not bind to JIP4 [22], supporting the lack of functional redundancy in Phafin family proteins.

## 6. Physiological and Pathological Functions of Phafins

### 6.1. Immunity

Phafin2 is a pattern recognition receptor in the innate immune system of zebrafish. Recombinant Phafin2 proteins bind to bacterial signature molecules in vitro, such as peptidoglycans, lipopolysaccharides, and lipoteichoic acid [28]. More importantly, Phafin2 inhibits the growth of both Gram-positive and Gram-negative bacteria [139]. It has been speculated that the antibacterial activity of Phafin2 may be critical for the early development of zebrafish embryos.

The *Phafin2* gene shows a high expression level in immune cells such as CD19+ B-cells and BDCA+ dendritic cells (http://biogps.org/#goto=genereport&id=79666 (accessed on 16 November 2022)), which suggests that Phafin2 may play a key role in the immune response [21,140]. Macropinocytosis has been receiving much attention because of its involvement in immune defense. Dendritic cells utilize macropinocytosis to sample their immediate environment for antigens. Therefore, given the role of Phafin2 in macropinocytosis, it is reasonable to propose that Phafin2 plays a role in cellular immune responses [21].

### 6.2. Cancer

The *Phafin2* gene was identified as an estrogen-responsive gene in breast cancer (BC) cells through DNA microarray analysis of gene expression profiles. The expression profile of the *Phafin2* gene in estrogen-responsive BC cells is significantly different from that in non-responsive cells. *Phafin2* is one of a set of 61 genes identified through in silico metanalysis of the transcriptomes of estrogen receptor-positive and estrogen receptor-negative human BC cell lines whose expression profiles discriminate these two cellular phenotypes in BC cell lines and tumor biopsies [141]. In addition, the *Phafin2* gene is differentially expressed in human hepatocellular carcinomas (HCC). The Phafin2 mRNA level is significantly higher in the transcriptome of HCC tumors than in normal livers, indicating that *Phafin2* gene expression is upregulated in liver cancer [14].

Both Phafin1- and Phafin2-expression levels were found to be upregulated in multiple cancers by screening the publicly available cBioportal cancer genome database [21]. In the case of the *Phafin2* gene, amplification is the most frequent type of gene alteration. The amplification frequency of Phafin2 in BC, prostate cancer, and bladder/urinary tract cancer is more than 15%. It has been proposed that overexpression of Phafin2 gives cancer cells a growth advantage, enhancing their nutrient scavenging, possibly through its involvement in macropinocytosis. Indeed, KRAS-transformed pancreatic cancer cells are not able to scavenge extracellular proteins under amino acid-limiting conditions and exhibit reduced proliferation after deletion of Phafin2 [21]. Even though the *Phafin1* gene is also amplified in various cancers [15,22], it remains unknown as to what the pathological consequences of such an amplification are.

## 7. Conclusions and Perspectives

Phafins are PH domain and FYVE domain-containing proteins that act as adaptor proteins in apoptotic, endocytic, and autophagic pathways (Table 2). Their membrane association is primarily mediated by the FYVE domain because of its high specificity for PtdIns(3)P and the enrichment of PtdIns(3)P on endosomal-lysosomal membranes. In most cases, the PH domain is not important for membrane association, as the deletion of PH domains in Phafins does not affect membrane localization. However, Phafin2 shows a coincidence detection mechanism for PtdIns(3)P and PtdIns(4)P during the two phases of macropinosome maturation, which requires the lipid-binding activities of both the PH and FYVE domains [21]. Phafins bind and recruit their binding partners, such as Akt and JIP4 (Table 1), to the cellular membranes. Unlike the Phafin2-Akt interaction [72], association of Phafins and their protein partners is predominantly mediated by PH domains, which is consistent with the general observation that FYVE domains are lipid-binding modules, whereas PH domains are more diverse in ligand recognition.

Extensive membrane remodeling, such as membrane trafficking and membrane fusion, is required for endocytic and autophagic pathways. Phafin2 strongly colocalizes with lysosomes and autophagosomes during autophagy [72]. Phafin2 also directly interacts with EEA1, facilitates endosome fusion [81], and recruits JIP4 to tubulating macropinosomes, promoting tubulation [22]. Taken together, it is tempting to hypothesize that Phafin2 induces changes in membrane shape, facilitating PtdIns(3)P-dependent organelle membrane remodeling.

Autoinhibition plays an important role in regulating protein function. Autoinhibitory mechanisms are found in many catalytic proteins (e.g., kinases) and non-catalytic proteins (e.g., adaptor proteins) [142,143]. We previously showed that the Phafin2 FYVE domain constitutively binds to PtdIns(3)P, whereas the PH domain’s PtdIns(3)P binding is intramolecularly inhibited by the polyD motif [16]. Autoinhibition of PtdIns(3)P binding is possible in Phafin2 because of the presence of a large amount of random coil (~40%) in its structure, which may provide the appropriate flexibility for the autoinhibited and PH domain-dependent active conformations. This observation is supported by data from autoinhibited proteins, which are enriched in intrinsic disorder regions, mostly inhibitory domains [144]. The negatively charged C-terminal polyD motif (eight aspartic acids and two serine residues) may electrostatically interact with a positively charged PtdIns(3)P-binding pocket in the PH domain, facilitating this mechanism. A shorter polyD motif is found in Phafin1 proteins, but it is currently unknown whether it has a regulatory function. Interestingly, a unique C-terminal tail in Phafin1 is required for its lysosomal targeting [15]. A relative of Phafin1/LAPF in *S. cerevisiae*, Pib2, stimulates TORC1 signaling and regulates lysosomal membrane permeabilization in response to ER membrane stress [145]. Sequence alignment of Pib2 and other FYVE domain-containing proteins shows that Phafin1, but not Phafin2, is its closest relative because they share a 13-amino-acid “tail motif” at their C-termini. Pib2 associates with vacuolar and endosomal limiting membranes in unstressed yeast cells through its FYVE domain and PtdIns(3)P. The conserved C-terminal tail motif plays a significant role in the regulation of lysosomal membrane permeabilization [145]. Therefore, despite Phafin1 and Phafin2 having common functional domains, differences in their C-termini may support yet-to-be-discovered unique cellular functions.

## Figures and Tables

**Figure 1 ijms-24-08096-f001:**
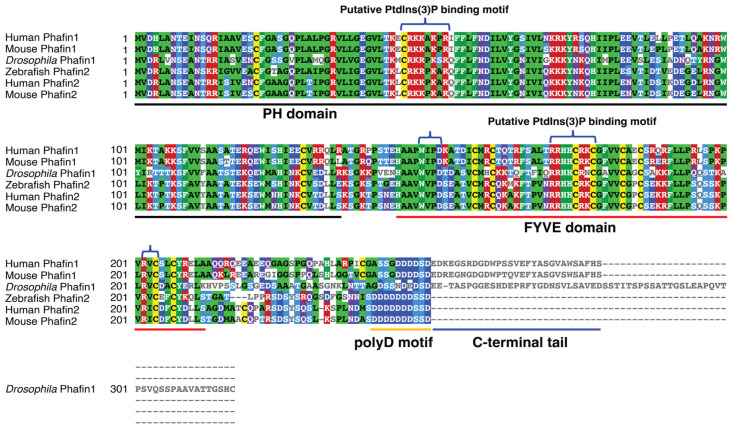
Sequence alignment of Phafin proteins. Amino acid sequences corresponding to *Homo sapiens* Phafin1 (UniProt accession number Q96S99), *Mus musculus* Phafin1 (UniProt accession number Q3TB82), *Drosophila melanogaster* Phafin1 (UniProt accession number O76902), *Danio rerio* Phafin2 (UniProt accession number Q7ZUV1), *Homo sapiens* Phafin2 (UniProt accession number Q9H8W4), and *Mus musculus* Phafin2 (UniProt accession number Q91WB4) were aligned using Clustal Omega. Sequence alignment is shown and colored using MView and structural and functional elements are underlined (PH domain, black; FYVE domain, red; polyD motif, yellow; and C-terminal tail, blue). The putative PtdIns(3)P-binding sites in the PH and FYVE domains are marked by braces. Of note, three conserved sequences, the N-terminal WxxD, the central RR/KHHCR, and the C-terminal RVC motifs in the FYVE domain form a compact PtdIns(3)P-binding site.

**Figure 2 ijms-24-08096-f002:**
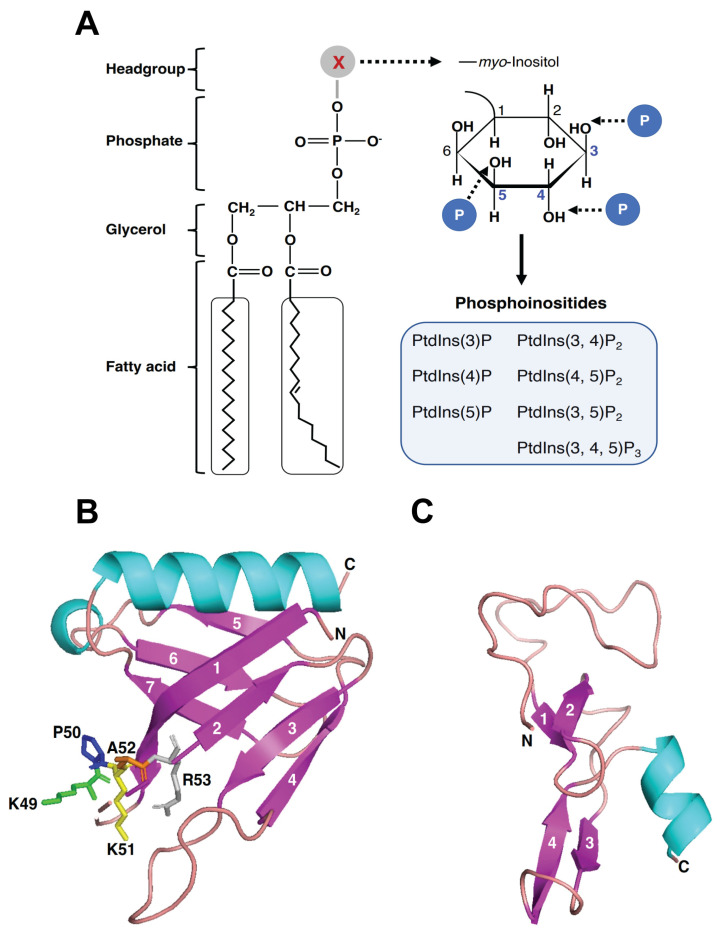
Phosphoinositides and potential PIP-binding sites in the Phafin2 PH and Phafin2 FYVE domains. (**A**) Molecular formulas of a glycerophospholipid and its seven derived PIPs. (**B**,**C**) Structure models of the human Phafin2 PH (**B**) and FYVE (**C**) domains generated by the SWISS-MODEL online program using templates with high sequence similarity (for PH domain, PDB accession code: 4gzu.1; for FYVE domain, PDB accession code: 3t7l.1.A). A basic sequence motif of 49-KPKAR-53 in the β1/β2 loop is shown in stick format, which corresponds to the canonical PIP-binding motif KXn(K/R)XR in PH domains.

**Figure 3 ijms-24-08096-f003:**
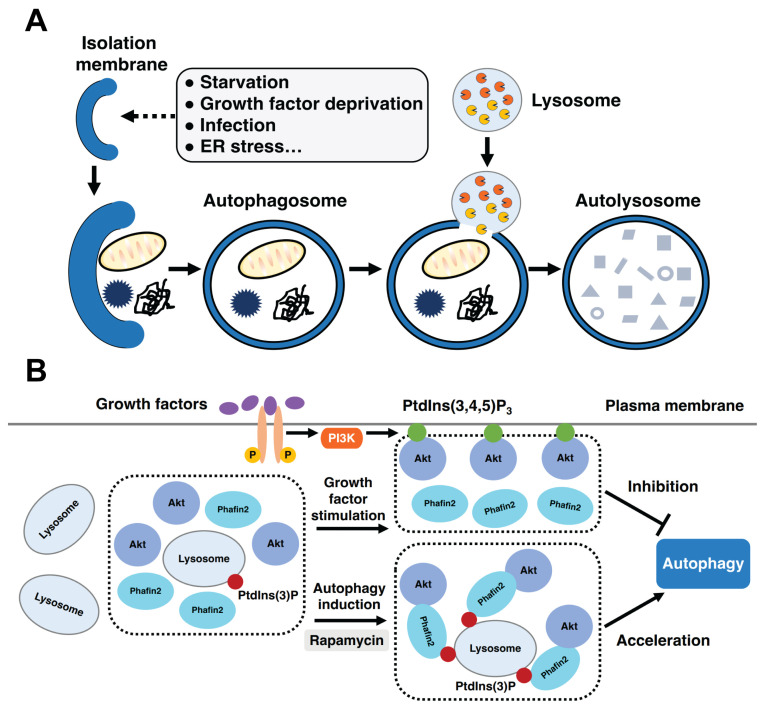
The autophagic pathway and the role of Phafin2 in autophagy. (**A**) Phases of the autophagic pathway. (**B**) The role of Phafin2, Akt, and PIP in autophagy.

**Figure 4 ijms-24-08096-f004:**
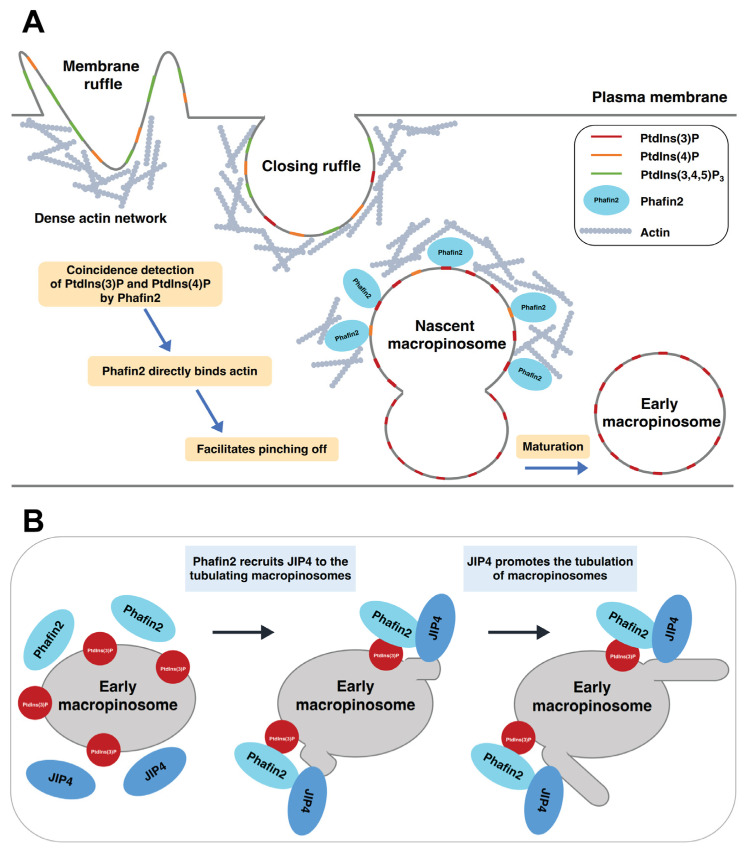
The role of Phafin2 in macropinocytosis. (**A**) The process of micropinocytosis. Phafin2 binds to PtdIns(3)P and PtdIns(4)P on nascent macropinosomes through a coincidence detection mechanism. Additionally, Phafin2 directly interacts with actin, shedding the dense actin cytoskeleton, and facilitating maturation of macropinosomes. (**B**) Phafin2 recruits JIP4 to the tubulating macropinosomes and promotes tubulation in a PIP-dependent manner.

**Table 1 ijms-24-08096-t001:** Binding partners of Phafin proteins.

Proteins	Binding Partners	Functional Domains	References
Phafin1	PtdIns(3)P	PH and FYVE domains	[15]
Phafin2	PtdIns(3)P, PtdIns(4)P	PH and FYVE domains	[21]
Rush *	GDI	-	[80]
Phafin2	EEA1	-	[81]
Phafin2	Akt	PH and FYVE domains	[72]
Phafin2	Actin	PH domain	[21]
Phafin2	JIP4	PH domain	[22]

* A homolog of human Phafin1 and human Phafin2 in *Drosophila melanogaster*. Abbreviations: GDI, guanosine diphosphate (GDP) dissociation inhibitor; EEA1, early endosomal antigen 1; Akt, protein kinase B; JIP4, JNK-interacting protein 4.

**Table 2 ijms-24-08096-t002:** The role of Phafins in cellular pathways.

Phafins	Cellular Functions	References
Rush *	Endosomal cargo transport	[80]
Phafin1	Autophagy	[15]
Phafin2	Macropinocytosis	[21]
	Apoptotic Pathways	[20]
	Endosomal cargo transport	[14,81]
	Autophagy	[72]

* A homolog of human Phafin1 and Phafin2 found in *Drosophila melanogaster*.

## Data Availability

No new data were created or analyzed in this study. Data sharing is not applicable to this article.

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
