# Peer review of "Phafins Are More Than Phosphoinositide-Binding Proteins"

_ijms, 2023, doi:10.3390/ijms24098096_

Round 1

Reviewer 1 Report

The authors review our current understanding of proteins of the phafin family characterized by the presence of a FYVE domain and a PH domain.

As general remark, I find the author insisting too much on the interaction of the PH domain with PI3P and should be more open throughout their review to the fact that PH domains can bind a number of different molecules phosphoinositides or proteins as they mention briefly lines 180-181. As far as I know, the true binding partner of the PH domain of phafins has not been described. Furthermore, I would make little sense that it binds PI3P when its neighboring domain is FYVE that is very specific to PI3P. I would rather that a more balanced view on this PH domain. Furthermore they cite in table 1 all the binding partners of phafins found so far and they are all proteins.

I would start the introduction with a sentence along the line of ' Cells are characterized my many membrane dynamics' and then describe what those are.

It is unclear that phafins are restricted to the Animal kingdom. the authors never discuss the fact that Mammals have 2, fish 1 of one family and drosophila 1 of the other family, what bearing could it have on its function.  This could be a good opportunity present a more detail genetic distribution of these proteins in the Animal kingdom. How many do birds have, do all fish have only phafin2 and do all invertebrates have phafin1.

On the same line, their presentation suggests that the 2 subfamilies are not present in the same cell lines. The authors should either clearly state whether they are or not, or not emphasize this aspect.

Line 203-205, they describe the FYVE domain as 2 double-stranded beta-sheets, each sheet constituted od 2 anti-parallel beta-strands. The figure shown shows only 1 beta-sheet.

Reviewer 2 Report

In their manuscript “Phafins are more than phosphoinositide-binding proteins.”, Tang et al review the current knowledge on Phafin proteins. This protein family – which contains two members, Phafin1 and Phafin2, bind to phosphoinositide lipids and regulate several cellular functions, including endocytic trafficking, autophagy, macropinocytosis and apoptosis.

The review is nicely written and gives a good summary of the current knowledge on Phafin proteins. It gives a nice overview of the different biological processes in which Phafin molecules play a role, and describe the roles of the Phafins in these processes.

There are some inaccuracies in some statements which should be addressed by the authors (see below). Especially the current view on the role of Phafins in apoptosis has been tainted by the retraction of a major paper describing this process, and it would be good to critically review the literature and maybe use the opportunity to discuss if the current knowledge without this study still supports this role of Phafins.

Besides these small criticisms, I fully support the publication of this review .

a)  Role of Phafins in apoptosis:

One of the main references (#14) describing a role of LAPF/Phafin1 in apoptosis has been retracted by the journal (https://pubmed.ncbi.nlm.nih.gov/16188880/) and should not be cited.

A second paper, describing a role of EAPF / Phafin2 in apoptosis (PMID 18288467 , citation #21) , has also remarks on Pubpeer, and should likely be treated with care.  There is a third paper, PMID 18056442  (citation #80) from the same group, which does not have any notes of concern so far; however, given that the same group publishing these three papers has more than 70 problematic papers listed on Pubpeer (https://forbetterscience.com/2021/06/02/the-16-secret-retractions-of-xuetao-cao-at-jbc/) , it might be wise to take these publications with a grain of salt.

b)  …”Phafin1 is not involved in micropinocytosis [22,23]”. The cited manuscripts do not test a role of Phafin1 in macropinocytosis, but only binding to Jip4 (in reference 23). Also note the spelling error in this sentence.

Reviewer 3 Report

The manuscript provides a comprehensive overview of Phafins, a family of adaptor proteins with PH and FYVE domain, involved in apoptotic, endocytic, and autophagic pathways. The authors have presented a well-written and structured manuscript that effectively summarizes the current knowledge about Phafins, their membrane association, and regulatory mechanisms. The review highlights the functional roles of Phafin proteins in various cellular pathways, such as apoptosis, endocytic cargo trafficking, and autophagy. Additionally, the review discusses the differences and similarities between Phafin1 and Phafin2. The overview of PH and FYVE domain and PIPs is thorough and provides a great background.

Overall, the review provides a solid introduction to the Phafin protein family and their functions in cellular pathways. It would be a useful resource for researchers interested in this field.

The authors may want to think about incorporating graphical representations to make the manuscript more readable. By including figures and tables, they can effectively summarize intricate information and enhance the clarity of the manuscript. Adding a single figure that summarizes the major known roles of Phafins would be immensely helpful. Additionally, including a table that summarizes the various functional roles and their corresponding references would be a valuable addition.
